# Contribution to the Understanding of the Interaction between a Polydopamine Molecular Imprint and a Protein Model: Ionic Strength and pH Effect Investigation

**DOI:** 10.3390/s21020619

**Published:** 2021-01-17

**Authors:** Amal Tlili, Ghada Attia, Sohayb Khaoulani, Zouhour Mazouz, Chouki Zerrouki, Nourdin Yaakoubi, Ali Othmane, Najla Fourati

**Affiliations:** 1LIMA Laboratory, Faculty of Medicine of Monastir, Monastir University, Av. Avicenne, Monastir 5019, Tunisia; amaltlili24@gmail.com; 2SATIE Laboratory, Cnam, UMR CNRS 8029, 292 Rue Saint Martin, 75003 Paris, France; ghada.attia@lecnam.net (G.A.); sohayb.khaoulani@lecnam.net (S.K.); zerrouki@cnam.fr (C.Z.); 3NANOMISENE Laboratory, CRMN, Technopôle Sousse, Sousse University, Sousse 4050, Tunisia; mazouz.zouhour645@gmail.com; 4LAUM Laboratory, Le Mans University, UMR CNR 6613, Avenue Olivier Messiaen, CEDEX 9, 72085 Le Mans, France; Nourdin.Yaakoubi@univ-lemans.fr

**Keywords:** molecularly imprinted polymer, bovine serum albumin, ionic strength, pH, dissociation constant

## Abstract

Several studies were devoted to the design of molecularly imprinted polymer (MIP)-based sensors for the detection of a given protein. Here, we bring elements that could contribute to the understanding of the interaction mechanism involved in the recognition of a protein by an imprint. For this purpose, a polydopamine (PDA)-MIP was designed for bovine serum albumin (BSA) recognition. Prior to BSA grafting, the gold surfaces were functionalized with mixed self-assembled monolayers of (MUDA)/(MHOH) (1/9, *v/v*). The MIP was then elaborated by dopamine electropolymerization and further extraction of BSA templates by incubating the electrode in proteinase K solution. Three complementary techniques, electrochemistry, zetametry, and Fourier-transform infrared spectrometry, were used to investigate pH and ionic strength effects on a MIP’s design and the further recognition process of the analytes by the imprints. Several MIPs were thus designed in acidic, neutral, and basic media and at various ionic strength values. Results indicate that the most appropriate conditions, to achieve a successful MIPs, were an ionic strength of 167 mM and a pH of 7.4. Sensitivity and dissociation constant of the designed sensor were of order of (3.36 ± 0.13) µA·cm^−2^·mg^−1^·mL and (8.56 ± 6.09) × 10^−11^ mg/mL, respectively.

## 1. Introduction

Molecularly imprinted polymers (MIPs) have received widespread attention in bioanalytical methods such as solid phase extraction (SPE) [1], catalytic processes [2], and chromatography techniques [3]. MIPs are increasingly investigated due to their numerous advantages: high affinity towards analytes of interest; chemical, physical, and thermal stability; and straightforward preparation [4]. The rational design of molecular imprinting consists in generating synthetic receptors mimicking natural recognition elements [5]. Concretely a polymeric matrix is first achieved around the molecules of interest. Various types of interactions can be involved, between the functional monomers, the analyte, and the cross linker, during this process, e.g., electrostatic forces, π–π interactions, hydrogen bondings, Van Der Waals forces, and hydrophobicity [6,7]. The further extraction of templates permits to release accessible sites for the analyte rebinding [8]. The recognition process is consequently controlled by the orientation, arrangements, and the nature of functional groups within these binding sites [5,9].

The success of small molecules imprinting has motivated the researchers to investigate more complex templates, such as viruses [10], peptides [11], and especially proteins [12]. Nevertheless, proteins’ molecular imprinting is complex because of: (i) the large size of these entities, which makes possible their deep tapering in the polymeric matrix and rends thus difficult templates release [13]; (ii) proteins dependence on osmolarity, ionic strength, and pH, three parameters, which can alter both their stability and their solubility [13]; and (iii) the possibility of protein’s conformation change during the polymerization process [14].

In a previous work, Bereli et al., have designed lysozyme (Lyz)-imprinted supermacroporous cryogels to purify Lyz protein from egg’s white [15]. The authors have explored the effects of external stimuli, such as salt concentration (NaCl), on the lysozyme adsorption. Their results indicate that the augmentation of NaCl concentration leads to a decrease in Lyz adsorption capacity from 16.6 to 6.7 mg/g. This was attributed to the repulsive electrostatic forces between the Lyz-MIP cryogel and lysozyme molecules.

Effects of ionic strength was also investigated by Gai et al., ref. [16], who have designed a bovine hemoglobin (BHb)-based molecularly imprinted gel polymers. Contrarily to Bereli et al., ref [15], their results indicate that hemoglobin adsorption is not dependent on NaCl concentration. The authors suppose that the electrostatic forces do not play, in this case, any role in the interaction between the investigated protein and the acrylamide monomers.

The effect of pH was investigated by Wang et al., who have synthetized tow MIPs for BSA [17] and BHb [18] detection. Results indicated that templates adsorption was maximum at pH 5.0 and 7.0 for BSA and BHb, respectively. This was attributed to the fact that these pH values were close to the isoelectric point (IP) of each protein. Saylan et al. [19] have studied the effect of pH on IgG adsorption in silica microsphere surface modified by aspartic acid MIP. They reported that the maximal adsorption of IgG is found at pH 6, near to IgG IP (6.2). They attributed these finding to the electrostatic interaction between the positive charge of IgG and negative charge of aspartic acid (IP = 2.77). Nevertheless, it was reported in other studies [20,21] that proteins were less soluble and less stable around their IP.

The inconsistency of these results raised the necessity to understand the effects of pH and ionic strength on the mechanisms of interaction between a considered protein and the imprinting sites. This is what we tried to do in this study. A polydopamine molecularly imprinted polymer was thus designed for BSA protein model detection. This choice was motivated by the fact that dopamine (DA) and BSA have comparable isoelectric points. In addition, DA is stable in different ionic strength and pH values and can be electropolymerized in physiologic environment [22]. The MIP design was inspired from the strategy of Tretjako et al. [23], which consists in creating well-ordered cavities. BSA templates were removed from the vicinity of the polymeric matrix by incubating the imprinted polymer in proteinase K solution. The corresponding nonmolecularly imprinted polymer, NIP, was prepared in the same conditions but without adding BSA templates.

Square wave voltammetry, zetametry, and Fourier-transform infrared spectrometry were investigated to control the surfaces’ modification and to evaluate the effects of pH and the ionic strength on the interaction between a polydopamine molecular imprint and BSA protein model.

## 2. Materials and Methods

### 2.1. Chemicals

Sulfuric acid (H_2_SO_4_, 95%), hydrogen peroxide (H_2_O_2_, 30%), mercaptohexanol (MHOH), acid mercaptoundecanoic (MUDA), bovine albumin serum (BSA), dopamine (DA), and proteinase K (pK) were purchased from Sigma Aldrich (Paris, France). N-hydroxysuccinimide (NHS), 1-ethyl-3-(3 dimethylaminopropyl)-carbodiimide hydrochloride (EDC), acetone, and ethanol were procured from Thermo Fisher (Artenay, France). A phosphate buffer solution (PBS) tablets were purchased from VWR (Strasbourg, France). Each tablet was dissolved in 100 mL of double ionized water (having a resistivity of 18.2 MΩ). All the chemicals were of analytical grade and were used as received without further purification.

### 2.2. Instrumentation

#### 2.2.1. Electrochemical Measurements

Cyclic voltammetry (CV) and square wave voltammetry (SWV) were performed with a CHI 1222C. Gold, platinum, and Ag/AgCl were used as the working, the auxiliary, and the reference electrodes, respectively. The following parameters were entered in all SWV measurements: increment = 4 mV, amplitude = 25 mV, frequency = 25 Hz, quiet time = 2 s, and sensitivity = 1 × 10^−3^.

The work electrodes (S = 40 mm^2^) were elaborated by the evaporation of 20 nm/150 nm of chromium/gold thin layers on the surface of Kapton^®^ polymers. Here, chromium was used as an intermediate layer to improve gold adhesion. All electrochemical results were expressed in terms of current density variation and calculated according to the equation:$$J=\frac{Current\;Intensity}{Electrode\prime s\;surface}.$$

#### 2.2.2. Fourier-Transform Infrared Spectroscopy (FTIR)

FTIR analyses were carried out with a Perkin Elmer “Spectrum two” spectrometer equipped with an attenuated total reflectance (ATR) cell. The spectra were collected at room temperature from 500 to 4000 cm^−1^ with a resolution of 2 cm^−1^. A background spectrum was recorded prior to each measurement and the further “background corrections” were systematically operated.

#### 2.2.3. Zetametry

Zeta potentials measurements were performed with a Malvern Zetasizer Nano ZS. For each analysis, 0.1 g of BSA protein was dissolved in 1 mL of PBS, and zeta potential value was obtained by averaging three measurements.

### 2.3. MIP Preparation

Various cleaning steps were carried out on the gold surfaces prior to the MIP realization: the substrates were first immersed during 5 min in acetone and then for 5 min in ethanol under plannar agitation. The electrodes’ surfaces were then activated with a piranha solution (95% H_2_SO_4_/30% H_2_O_2_1:1 *v/v*) during 10 min to favor the formation of OH groups on gold surfaces. This step was followed by a copious rinsing with ultrapure water and then with ethanol before drying under ambient air. After that, the substrates were incubated during 24 h, at 4 °C, in 3 mL of an ethanolic solution containing 0.1 M MUDA and 1 mM MHOH. A further ethanol rinsing (5 min) allowed the removal of the nonfixed molecules from the newly formed self-assembled monolayers (SAMs). A 50 µL drop of EDC/NHS (75 mM/15 mM) solution was then deposited on the modified electrode, at ambient temperature and during 30 min, to activate the -COOH groups. This step was followed by rinsing with ultrapure water prior to the deposition of a drop of 50 µL of 0.1 mg/mL of BSA dissolved in PBS at pH 7.4. The modified surfaces were then washed with PBS solution to eliminate the non adsorbed proteins. The following step consists in electropolymerizing 20 mM of dopamine, dissolved in 0.01 M PBS, by voltammetry cyclic (CV) at a potential ranging from −0.5 to 0.5 V and at a scanning speed of 50 mV/s. The optimum thickness of the polydopamine (PDA) film was obtained after 17 CV cycles (results not shown here). Further, 50 µL of 20 mg/mL of proteinase K solution was then deposited on the substrates, during 2 h at 40 °C to remove the BSA templates. The electrodes were finally copiously rinsed with PBS, during 15 min, and ultrapure water, during 15 min, to eliminate any traces of nonadsorbed BSA molecules. A schema of the MIP’s design strategy is presented in Figure 1a.

To control the affinity between the MIP and BSA protein, a nonimprinted polymer (NIP) was also designed in the same conditions as the MIP without adding BSA proteins, namely, MUDA/MHOH mixed SAMs were first grafted on the gold surfaces. MUDA’s COOH groups were then activated by EDC/NHS prior to dopamine electropolymerization. The designed NIPs have thus the same chemical properties as the MIP ones, but do not contain specific cavities of BSA protein.

## 3. Results and Discussion

### 3.1. MIP Design

#### 3.1.1. MIP and NIP Electrochemical Characterization

Square wave voltammetry was then conducted in PBS solutions, without adding redox probes, to control all the surface’s modifications steps (Figure 1a). The MIP and NIP voltammograms, before and after incubation in proteinase K solution, are depicted in Figure 1b. Here, both BSA and polydopamine are electroactive in the investigated potential range. In fact, BSA protein is riche in electroactive amino-acid groups such as tryptophane, cysteine, and tyrosine [24], while polydopamine possesses a high functional activity and semiconducting properties [22]. The similarities between the functional groups in both polydopamine and BSA protein is probably at the origin of the generation of an electrochemical response, recorded in the same potential range. Besides, the square wave voltammograms presented in Figure 1b, highlight a significant current density decrease and a potential shift (from 0.371 to 0.315 V) after BSA extraction. This indicates that the proteinase K has successfully removed the proteins from the vicinity of the polymeric matrix.

The percentage of extraction rate (*ER*) was calculated from the equation:(1)$$ER=\frac{J_{\mathrm{MIP}}-\;J_{\mathrm{extracted}\;\mathrm{MIP}}}{J_{\mathrm{MIP}}}\;\times \;100,$$
where *J*_MIP_ and *J*_extracted MIP_ are the current densities maxima of the MIP before and after the extraction step, respectively. *ER* was of order of 72%.

The square wave voltammograms of the NIP before and after incubation in proteinase K solution are presented in Figure 1b. Compared to the MIP, the NIP presents a lower variation of current density. This can be attributed to the compact structure of the NIP which may hinder the electrons transfer toward the electrode surface. The current peak decrease after NIP dipping in proteinase K solution is probably due to the elimination of self-polymerized polydopamine fragments that were not directly and strongly grafted to the electrode surface [25].

#### 3.1.2. FTIR Characterization

Fourier-transform infrared (FTIR) spectra of the MIP and the NIP are presented in Figure 2a. The similarity between these spectra is related to the fact that these polymers have the same backbone. For the large peak in the zone 1, ranging from 4000 to 3000 cm^−1^, we have chosen to operate a Gaussian deconvolution to better identify the functional groups present in the MIP and NIP structures (Figure 2b). The obtained bands at 3478 and 3303 cm^−1^ were assigned to the stretching vibrations of the amino N-H and hydroxyl O-H groups, respectively [26].

A zoom on the zone 2, ranging from 1900 to 800 cm^−1^, is presented in Figure 2c. The bands at 1737 and 1659 cm^−1^, observed for both the MIP and NIP spectra, can be assigned to the elongation vibration of the C=O group of the quinone, and the stretching vibration of the indole ring of polydopamine film, respectively [27]. Two specific bands are, however, present only for the MIP structure. The peaks at 1555 and 1468 cm^−1^, which can be attributed to the elongation vibration of the N−H and asymmetric elongation of C−N groups of secondary amide of BSA, respectively [28,29].

By comparing the FTIR spectra of MIP and NIP, we noticed that the MIP intensity is higher than the NIP one. This can be attributed to the creation of hydrogen bonds and electrostatic interactions [30,31] between the BSA protein and the polydopamine matrix.

According to FTIR results and because of the importance of hydrogen bonds and electrostatic interactions on MIP-protein interaction, we have thus decided to conduct an exploratory study to elucidate the ionic strength and pH effects on the interaction mechanism between a given molecularly imprinted polymer (the PDA film in our case) and the BSA protein model.

#### 3.1.3. Electrochemical MIP Characterization at Different Ionic Strength and pH Values

##### Ionic Strength Effects on MIP Elaboration

Studies, conducted to investigate the effect of high NaCl concentration on the MIP-protein interactions, have shown that this causes a decrease in proteins adsorption in their imprint’s [15,19]. Literature has also reported that an increase in NaCl, KCl, LiCl, or CsCl concentrations, induced denaturation and precipitation of proteins [32,33,34]. We have thus made the choice to use a phosphate buffer solution, which contains several ions NaCl, KCl, ${\mathrm{Na}}_{2}{\mathrm{HPO}}_{4}$ and $K_{2}{\mathrm{HPO}}_{4}$ and adjusted its ionic strength to 16.7, 42, 167, 250, and 334 mM at a fixed pH value of 7.4. This permits to investigate ionic strength effects in line of the buffering capacity of the considered solutions.

Square wave voltammetry was then investigated to follow-up the electrochemical responses of each designed MIP (results are not shown here). For each curve, we noted the maximum value of current density and, subsequently, plotted the variations of current densities versus ionic strength values on Figure 3.

Results presented in Figure 3 show that an increase in the ionic strength, during MIP construction, leads to current density augmentation and that the saturation is reached at 167 mM. This is attributed to the “salting-out effect” obtained at high salt concentrations leading to low protein stability [35].

##### pH Effects on MIP Design

pH can also be considered as a crucial key to the successful MIP-Protein design. EL-Sharif et al. have compared the recognition process between different proteins and their MIPs in various pH buffers [36]. They showed that depending on the isoelectric point of the investigated protein, the pH variations can influence templates/imprints specific binding. Based on the isoelectric point of the BSA (IP = 4.6), we selected, therefore, three pH values, i.e., 3.5, 7.4, and 9. During our investigations, we have varied the pH of the supporting electrolyte (pH_PBS_) and fixed the pH of the BSA templates (pH_BSA_), and then performed the inverse operation, i.e., fixing pH_PBS_ and varying pH_BSA_.

Here also, we have investigated square wave voltammetry to follow-up the electrochemical responses of each designed MIP, noted the maximum value of current density, and then plotted the variations of current densities versus pH values. For all these experiments, we have fixed both the ionic strength (I = 167 mM) and the BSA concentration (10^−1^ mg/mL). Results, presented in Figure 4a, show that the MIP current density increases according to the pH of the supporting electrolyte and of BSA, except for the couple (pH_BSA_ = 9 and pH_PBS_ = 9).

Such finding could be explained by: (i) the polydopamine became more richer by negatively charged groups, such as the quinone-imine and catechol ones [37]. These chemical groups could in fact improve the polydopamine conductivity and electroactivity [22] and (ii) at pH values higher than BSA IP, the protein will have a net negative charge, leading to its electroactivity enhancement, which is confirmed by the potential zeta measurement (Table 1). Consequently, the amount of immobilized BSA can be expected to increase.

To explain the particular shape of the curve corresponding to pH_BSA_ = 9 in Figure 4a, we have designed three MIPs with a fixed pH for the protein solution (pH_BSA_ = 9) and three different pH values for the PBS media: 3.5, 7.4, and 9. The corresponding square wave voltammograms, presented in Figure 4b, show three oxidation peaks, well separated and well resolved at only pH_BSA_ = 9. These peaks can be attributed to the three amino acid residues of BSA: cysteine, tryptophan, and tyrosine [24,38]. Figure 4b shows also that the current density of the MIP designed with (pH_BSA_ = 9 and pH_PBS_ = 9) is inferior to the other ones, which may explain the particular variation in Figure 4a.

These results stressed the fact that MIP elaboration is greatly dependent on the values of pH and ionic strength of both the templates and the supporting electrolyte, but what about the effects of these parameters on the analyte detection after the extraction procedure?

### 3.2. Effects of Ionic Strength and pH on BSA Detection

#### 3.2.1. Effects of Ionic Strength

Five MIPs were first prepared at a fixed pH of 7.4 at different ionic strengths (16.7, 42, 167, 225, and 334 mM), and then dipped in proteinase K solutions during 2 h to remove the BSA templates. After that, the designed MIPs were incubated in BSA solutions with concentrations increasing from 10^−15^ to 10^−1^ mg/mL. Square wave electrochemical characterizations were then performed in PBS solutions prepared at the ionic strength of interest. For each measurement, we have noted the current density of BSA oxidation peak, and then plotted, for the whole measurements, the maximum of current densities versus BSA concentrations (Figure 5a). Results indicate that the maximum current density of BSA oxidation peak increases with ionic strength from 16.7 to 167 mM, and that beyond this value, an ionic strength increase leads to a current density decrease. These results can be explained by the fact that salt molecules can modify the MIP structure enhancing thus its volume, confirming thus the findings of Chen et al. in their study concerning lysozyme protein binding in a hydrogel layer [39]. Moreover, and according to Wang et al. [17,18], salt ions can also increase the proteins hydrability, which minimizes the protein affinity towards their polymeric cavities.

Zetametry technique was then performed to estimate zeta potential values of BSA protein under different salt concentrations (Figure 5b). Results indicate that, up to 167 mM, an ionic strength increase leads to a zeta potential decrease probably due to protein-protein repulsive interactions [40] and the good solubility and stability of BSA [41,42]. On the contrary, high ionic strengths provoke a slight augmentation of zeta potentials because of the consolidation of protein–protein interactions, thereby minimizing BSA solubility [32,35]. These results confirm that at high salt concentration, the hydrophobic regions of the protein react together and induce protein aggregation [35,43].

#### 3.2.2. Effects of pH

For a better understanding of the interaction between BSA and the polydopamine film, at different pH values (3.5, 7.4, and 9) and at a fixed ionic strength of 167 mM, we have compared the recognition of BSA analytes by both the imprinted and nonimprinted polymers. Here, we have made the choice to represent Δ*J*_MIP_ = *J*_MIP_ − *J*_MIP_ext_ and Δ*J*_NIP_ = *J*_NIP_ − *J*_NIP_ext_ versus cumulative BSA concentrations, where *J*_MIP_, *J*_NIP_, *J*_MIP_ext_, and *J*_NIP_ext_ represent the recorded current densities of the MIP, the NIP, the extracted MIP, and the NIP dipped in the proteinase K extraction solution, respectively. Experiments were done three times for each couple of pH values (pH_BSA_ and pH_PBS_), and we have considered the average curve as the calibration one. The obtained results are gathered on Figure 6.

To highlight the pH effect and for a better visibility, we considered separately the cases where the pH_BSA_ and/or pH_PBS_ values are lower than the BSA isoelectric point (IP), of order of 4.6 according to our zetametry measurement (Figure 6a), and the cases where both pH values are higher than IP of BSA (Figure 6b).

Results presented in Figure 6a indicate that the MIP and NIP responses can be considered as similar for the first three pH pairs, given the measurement’s dispersions and their associated uncertainties. This means that under these pH conditions, the adsorption phenomenon is relatively greater than the specific recognition of the analytes by the imprints. The rearrangement of surfacic charges and the variation of their density are probably at the origin of this unspecific recognition. Indeed, the polydopamine and the BSA are positively charged at pH lower than their isoelectric points, of order of 4 [44] and 4.6, respectively. This favors repulsive interaction and explains the fact that both NIP and MIP responses are the lowest ones when the pH values of BSA and PBS are equal to 3.5, and especially at higher concentrations of BSA.

When the pH values of BSA and PBS are higher than the isoelectric point of the BSA, the responses of the NIP and the MIP can be perfectly discriminated (Figure 6b), even when taking into account the results’ dispersion and their uncertainties. This means that under these pH conditions, the BSA imprints play their expected recognition role.

Prior to the modelling of the electrochemical response of the MIP-based sensor, we have modelled the NIP-based one to remove the contribution of nonspecific adsorption response. Although it was not possible to obtain a unique model for all considered measurements, because of the great dispersion of results, we nevertheless noticed that a power law was suitable in the majority of the cases. This allows us to consider a combination of two models to fit the MIPs electrochemical responses: a “one site binding,” for specific recognition, and a “power-law,” for the nonspecific one (Equation (2)):(2)$$\Delta J_{\mathrm{MIP}}\left(C\right)=\left(a+b\times C^{\alpha }\right)+\frac{m\times C}{K_{d}+C},$$
where Δ*J*_MIP_(*C*) is the current density variation (*J*_MIP_(*C*) − *J*_MIP-ext_(*C*)) for a given concentration *C* of BSA, *K*_d_ is the dissociation constant, *a*, *b*, *m*, and the ponderation exponent α are empiric constants.

In Figure 7 and its further inset are presented the calibration curves and their corresponding fits, in both linear and semilogarithmic scales. Sensitivities of the designed sensors were calculated from the semilogarithmic variations of the current densities, as they are quasilinear in a large concentration interval [10^−15^, 10^−3^] mg/mL. The calculated sensitivity and dissociation constants values are gathered in Table 2.

Table 2 data indicate that the best sensing condition, in terms of dissociation constant and sensitivity values, is that corresponding to (pH_BSA_ = 7.4 and pH_PBS_ = 7.4). For the couples (pH_BSA_ = 7.4 and pH_PBS_ = 9) and (pH_BSA_ = 9 and pH_PBS_ = 7.4), the reading depends on whether the sensitivity or the dissociation constant is considered as the relevant parameter. Notice that the configuration in which both considered pH values are equal to 9 is the least favorable one and should not be considered in proteins detection. The high values of the dissociation constants uncertainties are mainly due to the large variability of nonspecific recognition, as we have showed in the experiments involving the nonimprinted polymer.

To better understand the interaction mechanism between BSA protein and its imprint in different pH media, acid, neutral, and basic, FTIR studies have been conducted and are presented in the next section.

### 3.3. FTIR Analysis

It is evident that the variation of pH plays a crucial role in the interaction mechanism between the polydopamine film and BSA protein. Three MIPs were thus designed and then characterized by FTIR spectroscopy. Herein, we have adjusted the pH of the supporting electrolyte (pH_PBS_) and the pH of the BSA solution (pH_BSA_) at (3.5/3.5), (7.4/7.4), and (9/9). The corresponding spectra are presented in Figure 8.

Results show a variation in the band intensities at pH 3.5 in the zone 1, ranging from 3500 to 3300 cm^−1^. This can be attributed to the asymmetric valence vibrations of the ammonium ion (${\mathrm{NH}}_{3}^{+}$) formed during the (R-${\mathrm{NH}}_{2}$) protonation by the influence of the hydronium cation $\left(H_{3}O^{+}\right)$. At pH 7.4 and 9, these bonds are large and present a higher absorption intensity, which is probably due to different intermolecular hydrogen bonds. The peaks recorded in the zone 2 (1300–1200 cm^−1^) and around 1720 cm^−1^, for the MIP designed at pH 3.5, confirm the modification of the secondary structure of the protein, which leads to the appearance of the β-sheet conformation [45].

FTIR results indicate that the BSA is probably flattened at low pH values [46]. This may confirm our explanation in Section 3.2.1. We believe that the imprinted binding sites were not created at low pH values and that in this case, the interaction, between BSA and polymeric films, was only controlled by the electrostatic interaction.

## 4. Conclusions

We attempt in this work to understand the interaction mechanism between a protein model (BSA) and a polydopamine molecularly imprinted polymer. Three complementary techniques, electrochemistry, zetametry, and Fourier-transfer infrared spectroscopy, were used to understand the interaction between the PDA-MIP and BSA analytes and to investigate the roles of the ionic strength and the pH of both BSA and the buffer media. Results indicate that the best conditions, for a successful MIP-BSA design, were an ionic strength of 167 mM and a pH value of 7.4, two conditions that ensure a good structural stability of the protein and high electroactivity of the polydopamine polymeric matrix. The corresponding electrochemical sensor has a sensitivity of (3.36 ± 0.13) µA·cm^−2^·mg^−1^·mL and a dissociation constant K_d_ of order of (8.56 ± 6.09) × 10^−11^ mg/mL, which indicates a high affinity between the BSA protein and the polydopamine molecular imprint.

The final objective of this exploratory study is to draw attention on the fact that beyond the choice of the chemical structure of the functional monomers, the cross-linker agent and the polymerization technique, three other parameters are crucial in the design of a molecularly imprinted polymer that are the ionic strength, pH, and the isoelectric point. These parameters are of prime importance in the case of MIP designed for protein detection, as each protein in cells has different folding mechanisms and 3D structures in various pH and ionic strength conditions.

## Figures and Tables

**Figure 1 sensors-21-00619-f001:**
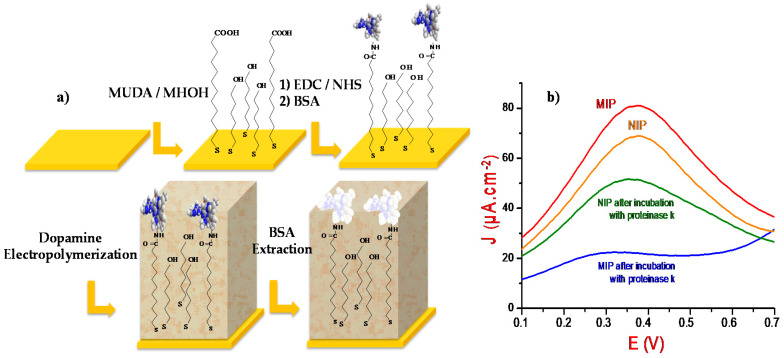
(**a**) Schematic representation of molecularly imprinted polymer (MIP) design strategy. (**b**) Square wave voltammograms of the MIP and the nonimprinted polymer (NIP) before and after incubation in proteinase K solution. Measurements were made in a 0.1 M phosphate buffer solution (PBS) solution.

**Figure 2 sensors-21-00619-f002:**
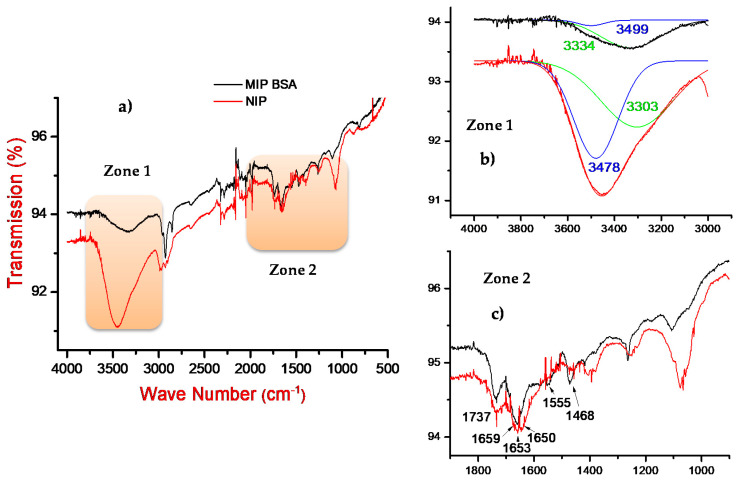
FTIR spectra of bovine serum albumin (BSA)/MIP and NIP at pH 7.4 (**a**) from 4000 to 500 cm^−1^. (**b**) A zoom on the zone 1 ranging from 4000 to 3000 cm^−1^ and the corresponding Gaussian deconvolution. (**c**) A zoom on the zone 2 ranging from 1900 to 800 cm^−1^.

**Figure 3 sensors-21-00619-f003:**
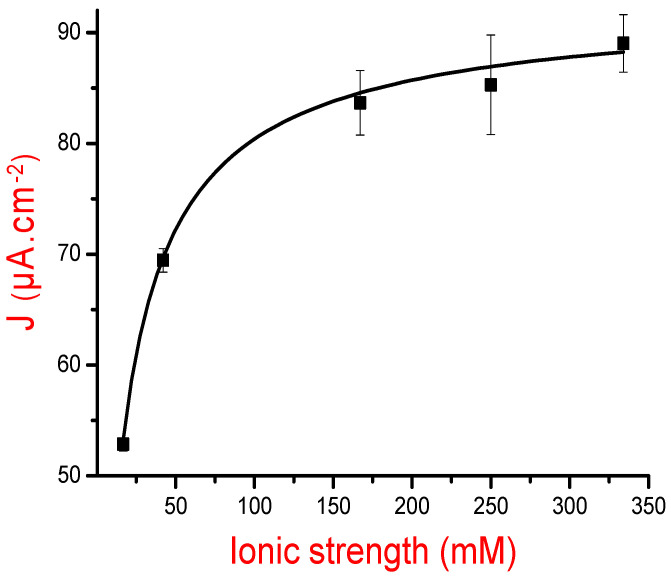
Variations of the current density maxima of MIPs designed at pH = 7.4 and varied ionic strengths versus the ionic strength of the BSA solution used in the MIPs’ design.

**Figure 4 sensors-21-00619-f004:**
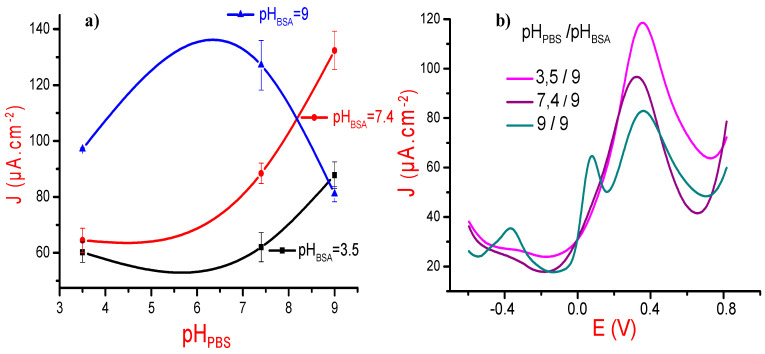
(**a**) Variations of the maxima of MIP current density versus the pH of the supporting electrolyte (pH_PBS_). pH values of BSA solutions used during the realization of the MIPs were equal to 3.5 (black curve), 7.4 (red curve), and 9.4 (blue curve). (**b**) Square wave voltammetry (SWV) voltammograms of MIPs designed with BSA protein solution at a fixed pH (pH_BSA_ = 9) and characterized at three different pH values of the PBS solution (pH_PBS_ = 3.5, 7.4, and 9). All experiments were done at a fixed ionic strength (I = 167 mM) and at BSA concentration equal to 10^−1^ mg/mL.

**Figure 5 sensors-21-00619-f005:**
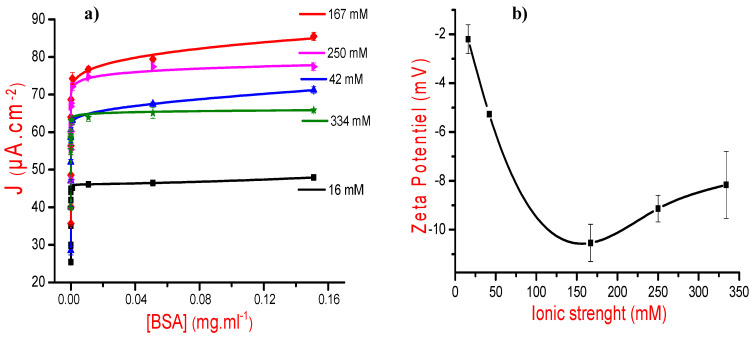
(**a**) Current density variations versus cumulative BSA concentrations for the four investigated ionic strengths: 16, 7, 42, 167, 250, and 334 mM (the lines are guides for the eyes). (**b**) Variation of the zeta potential versus the ionic strength (the lines are guides for the eye).

**Figure 6 sensors-21-00619-f006:**
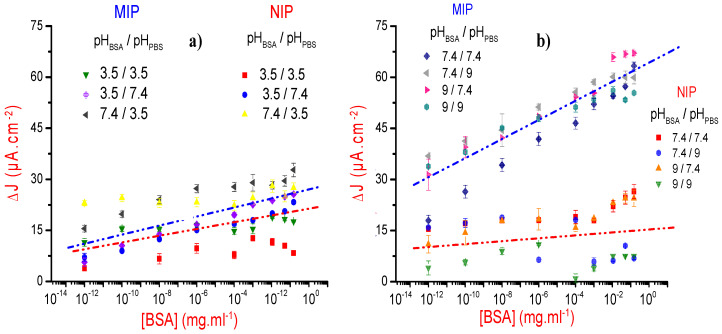
Variations of the current densities, Δ*J*_MIP_ and Δ*J*_NIP_ versus cumulative BSA concentration (in semilogarithmic representation): (**a**) For (pH_BSA_ or pH_PBS_) lower than BSA isoelectric point (IP). (**b**) For (pH_BSA_ and pH_PBS_) higher than BSA IP.

**Figure 7 sensors-21-00619-f007:**
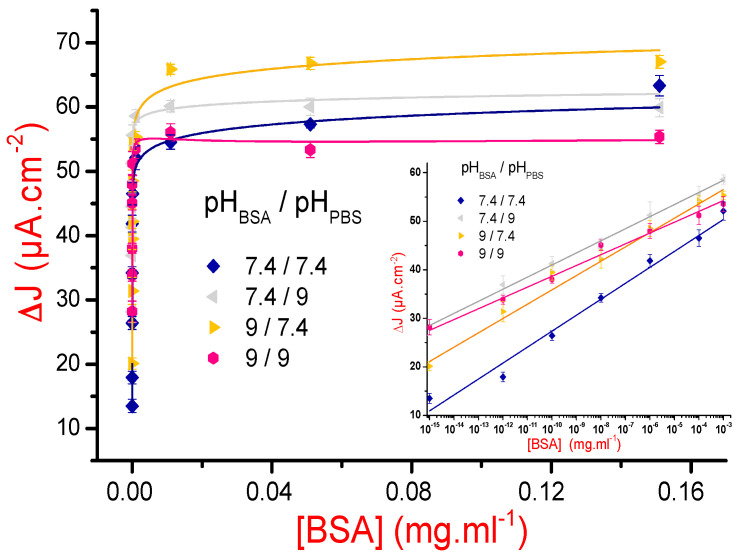
Variations of the current densities Δ*J*_MIP_ versus cumulative BSA concentration for pH_BSA_ and pH_PBS_ superior to the BSA isoelectric point. The solid lines correspond to the best fits according to Equation (2). Inset: semilogarithmic scale view showing the quasilinear shape of the calibration curves for BSA concentrations varying from 10^−15^ to 10^−3^ mg/mL.

**Figure 8 sensors-21-00619-f008:**
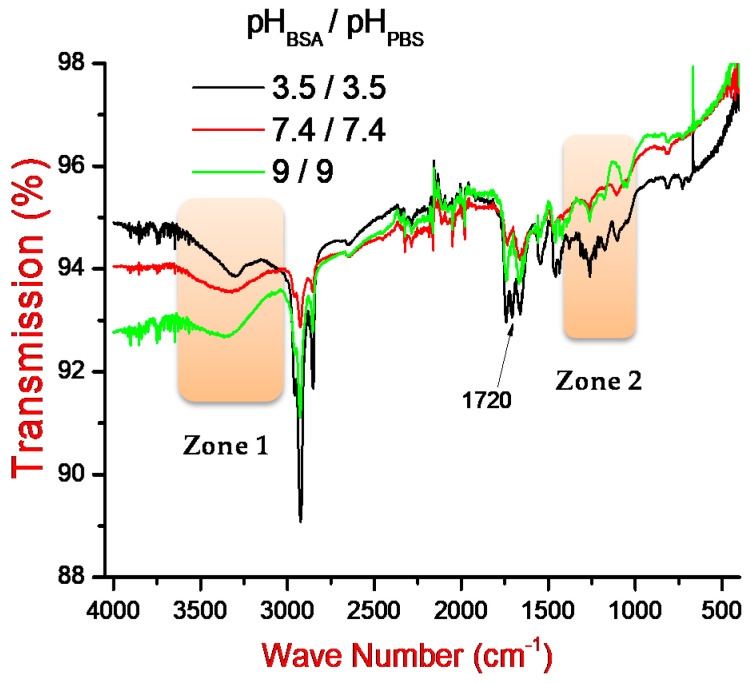
FTIR spectra of MIP response at pH_BSA_/pH_PBS_ adjusted to (3.5/3.5), (7.4/7.4) and (9/9).

**Table 1 sensors-21-00619-t001:** Zeta potential values of bovine serum albumin (BSA) solutions for which the pH was adjusted at 3.5, 7.4, and 9. The BSA protein was dissolved at a fixed concentration of 10^−1^ mg/mL.

pH_BSA_	Zeta Potential (mV)
3.5	5.77 ± 1.00
7.4	−11.11 ± 0.20
9	−10.66 ± 0.05

**Table 2 sensors-21-00619-t002:** Sensitivity and dissociation constants values.

pH_BSA_	pH_PBS_	Sensitivity (µA·cm^−2^·mg^−1^·mL)	Dissociation Constant (mg/mL)
7.4	7.4	3.36 ± 0.13	(8.56 ± 6.09) × 10^−11^
7.4	9	2.48 ± 0.08	(1.18 ± 1.38) × 10^−6^
9	7.4	3.01 ± 0.19	(0.75 ± 1.68) × 10^−6^
9	9	2.15 ± 0.08	(18.43 ± 32.45) × 10^−3^

## Data Availability

Not applicable.

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
