# Peer review of "Contribution to the Understanding of the Interaction between a Polydopamine Molecular Imprint and a Protein Model: Ionic Strength and pH Effect Investigation"

_sensors, 2021, doi:10.3390/s21020619_

Round 1

Reviewer 1 Report

The paper “Contribution to the comprehension of the interaction between a polydopamine molecular imprint and a protein model: ionic strength and pH effect investigation” by Amal Tlili, Ghada Attia, Sohayb Khaoulani, Zouhour Mazouz, Chouki Zerrouki, Nourdin Yaakoubi, Ali Othmane, and Najla Fourati describes interesting phenomena of bioanalytics. Imprinting is increasingly applied for the detection biopolymers. In this case surface imprinting has to be used. Anchoring procedures on gold electrodes are performed. Dopamine is electrochemically polymerized around the analyte. Detection methods are based on cyclic and voltammetry and square wave voltammetry.
It is obvious that ionic strength and pH variations will influence protein adsorption. Different strategies were applied to elucidate these phenomena such as electrochemistry, FT-IR and zetametry. The results are of fundamental importance for bioanalytics. It should be mentioned, however, that ionic strength introduced by Debye - Hückel theory describes more continuum phenomena. It would be interest that more specific ionic effects are observed at higher ion concentrations if you go perhaps from Li-ions to Cs-ions?

Author Response

Responses to reviewer 1 suggestions and comments

The authors would like to thank the reviewer for the interest that he paid for their work. His suggestion will permit to improve the quality of the manuscript.

Here after the response to reviewer’s question.

We agree that the ionic strength, and thus the nature and concentration of the investigated ions, affects both the stability and conformation of a given protein. The following paragraphs and some references were added in blue color and yellow highlighted to the text of the manuscript:

L204-210: “Studies, conducted to investigate the effect of high NaCl concentration on the MIP-protein interactions, have shown that this causes a decrease of proteins adsorption in their imprint’s [15,19]. Literature has also reported that an increase of NaCl, KCl, LiCl or CsCl concentrations, induced denaturation and precipitation of proteins [32-34]. We have thus made the choice to use a phosphate buffer solution, which contains several ions NaCl, KCl, Na2HPO4 and K2HPO4, and adjusted its ionic strength to 16.7, 42, 167, 250 and 334 mM at a fixed pH value of 7.4. This permits to investigate ionic strength effects in line of the buffering capacity of the considered solutions.”

L287-288: These results confirm that at high salt concentration, the hydrophobic regions of the protein react together and induce protein aggregation [35,44].

15. Bereli, N.; et al. Protein recognition via ion-coordinated molecularly imprinted supermacroporous cryogels. J. Chromatogr. A. 2008, 1190, 18–26.

19. Saylan, Y.; Üzek, R.; Uzun, L.; Denizli, A. Surface imprinting approach for preparing specific adsorbent for IgG separation. J. Biomater. Sci. Polym. Ed. 2014, 25, 881–894.

32. Parsons, D. F.; Boström, M.; Nostro, P. Lo; Ninham, B. W. Hofmeister effects: Interplay of hydration, nonelectrostatic potentials, and ion size. Phys. Chem. Chem. Phys. 2011, 13, 12352–12367.

33. Curtis, R. A.; Lue, L. A molecular approach to bioseparations: Protein-protein and protein-salt interactions. Chem. Eng. Sci. 2006, 61, 907–923.

34. Moghaddam, S. Z.; Thormann, E. The Hofmeister series: Specific ion effects in aqueous polymer solutions. J. Colloid Interface Sci. 2019, 555, 615–635.

35. Gerzhova, A.; Mondor, M.; Benali, M.; Aider, M. Study of total dry matter and protein extraction from canola meal as affected by the pH, salt addition and use of zeta-potential/turbidimetry analysis to optimize the extraction conditions. Food Chem. 2016, 201, 243–252.

44. Patel, R.; Kumari, M.; Khan, A. B. Recent advances in the applications of ionic liquids in protein stability and activity: A review. Appl. Biochem. Biotechnol. 2014, 172, 3701–3720.

Reviewer 2 Report

Comments:

  1. The current manuscript is well designed. However, the statistical analyses should be done and covered all the data on Figures and Tables. Authors only used BSA as an example for protein analysis. Each protein in cells have different folding mechanisms and 3D structures in various pH conditions. Thus, testing other proteins including some enzymes and carrier proteins would enhance the quality of the current manuscript. The current manuscript is in between "minor revision" and "major revision". I chose "major revision" is to ask authors re-do some statistical analyses and testing other proteins in addition to BSA unless authors can provide acceptable explanation.
  2. Statistical analyses should be done on Table 3 and other figures.

Author Response

Responses to reviewer 2 suggestions and comments

We thank the reviewer for all relevant comments and questions. His suggestions will permit to improve the quality of the manuscript.

Here after the point per point responses to the reviewer’s questions. All these elements were reported in the revised manuscript and indicated in blue color and yellow highlighted. Some references were also added.

     1. Testing other proteins in addition to BSA 

We fully agree that each protein has different folding mechanisms and 3D structures in various pH and ionic strength conditions and that investigating other proteins will enhance the quality of the manuscript.

It is this "uniqueness" that led us to present a single protein model to draw attention to the fact that, like the interactions between enzymes and their further substrats, the interactions between a MIP and a given protein are very strongly dependent on pH and ionic strength values. We have also tried to draw the attention of future readers to the need to understand the nature of the chemical bonds that can be created between a MIP and a given protein, which are extremely dependent on the isoelectric point of both the protein and the polymeric matrix.

The following paragraph was added to the conclusion:

“The final objective of this exploratory study is to draw attention on the fact that beyond the choice of the chemical structure of the functional monomers, the crosslinker agent and the polymerization technique, three other parameters are crucial in the design of a molecularly imprinted polymer that are the ionic strength, pH and the isoelectric point. These parameters are of prime importance in the case of MIP designed for protein detection, as each protein in cells has different folding mechanisms and 3D structures in various pH and ionic strength conditions.”

We count on the reviewer's understanding, and we hope to have convinced him that our approach, which consists in the study of a unique model protein, is exploratory, and that the findings can be “extrapolated” to other proteins.

     2. Statistical analyses should be done on Table 3 (table 1) and other figures.

The missed standard deviations were added in Table 1 and Figure. 6.

Reviewer 3 Report

This is a very well elaborated contribution. The experimental work was done with appropriate methods and was performed professionally. The findings are important and the interpretation is relevant. The manuscript is written well with good English, nevertheless, there is room for a final linguistic check-up.

Author Response

Responses to reviewer 3 suggestions and comments

The authors would like to thank the reviewer for the interest that he paid for their work. A final linguistic check-up has been done to improve the quality of the manuscript. Some sentences, paragraphs and references were added in the text to answer reviews’ questions and to make it easier for the readers. All modifications were indicated in blue color and yellow highlighted.

L72-76: Saylan et al. [19] have studied the effect of pH on IgG adsorption in silica microsphere surface modified by aspartic acid MIP. They reported that the maximal adsorption of IgG is found at pH 6, near to IgG IP (6.2). They attributed these finding to the electrostatic interaction between the positive charge of IgG and negative charge of aspartic acid (IP = 2.77).

L198 – 199: According to FTIR results and because of the importance of hydrogen bonds and electrostatic interactions on MIP-protein interaction …

L204-210: Studies, conducted to investigate the effect of high NaCl concentration on the MIP-protein interactions, have shown that this causes a decrease of proteins adsorption in their imprint’s [15,19]. Literature has also reported that an increase of NaCl, KCl, LiCl or CsCl concentrations, induced denaturation and precipitation of proteins [32,33,34]. We have thus made the choice to use a phosphate buffer solution, which contains several ions NaCl, KCl, Na2HPO4 and K2HPO4, and adjusted its ionic strength to 16.7, 42, 167, 250 and 334 mM at a fixed pH value of 7.4. This permits to investigate ionic strength effects in line of the buffering capacity of the considered solutions.

L222 – 226: pH can also be considered as a crucial key to the successful MIP-Protein design. EL-Sharif et al. have compared the recognition process between different proteins and their MIPs in various pH buffers [36]. They showed that depending on the isoelectric point of the investigated protein, the pH variations can influence templates/imprints specific binding. Based on the isoelectric point of the BSA (IP=4.6), we selected therefore three pH values: 3.5, 7.4 and 9.

L287-288: These results confirm that at high salt concentration, the hydrophobic regions of the protein react together and induce protein aggregation [35,44].

L397-402: The final objective of this exploratory study is to draw attention on the fact that beyond the choice of the chemical structure of the functional monomers, the crosslinker agent and the polymerization technique, three other parameters are crucial in the design of a molecularly imprinted polymer that are the ionic strength, pH and the isoelectric point. These parameters are of prime importance in the case of MIP designed for protein detection, as each protein in cells has different folding mechanisms and 3D structures in various pH and ionic strength conditions.

19. Saylan, Y.; Üzek, R.; Uzun, L.; Denizli, A. Surface imprinting approach for preparing specific adsorbent for IgG separation. J. Biomater. Sci. Polym. Ed. 2014, 25, 881–894.

32. Parsons, D. F.; Boström, M.; Nostro, P. Lo; Ninham, B. W. Hofmeister effects: Interplay of hydration, nonelectrostatic potentials, and ion size. Phys. Chem. Chem. Phys. 2011, 13, 12352–12367.

33. Curtis, R. A.; Lue, L. A molecular approach to bioseparations: Protein-protein and protein-salt interactions. Chem. Eng. Sci. 2006, 61, 907–923.

34. Moghaddam, S. Z.; Thormann, E. The Hofmeister series: Specific ion effects in aqueous polymer solutions. J. Colloid Interface Sci. 2019, 555, 615–635.

36. EL-Sharif, H. F.; Phan, Q. T.; Reddy, S. M. Enhanced selectivity of hydrogel-based molecularly imprinted polymers (HydroMIPs) following buffer conditioning. Anal. Chim. Acta. 2014, 809, 155–161.

44. Patel, R.; Kumari, M.; Khan, A. B. Recent advances in the applications of ionic liquids in protein stability and activity: A review. Appl. Biochem. Biotechnol. 2014, 172, 3701–3720.

Round 2

Reviewer 2 Report

No more comments

Author Response

Answers to the reviewers’ comments

The authors would like to thank the reviewer for the interest that he paid for their work. A final linguistic check-up has been done to improve the quality of the manuscript. Figures 3, 4 and 6 captions were changed. Some paragraphs were also added in the text to answer editors’ questions and to make it easier for the readers. All modifications were indicated in blue color and yellow highlighted in the manuscript.